# Precise Design Strategies of Nanotechnologies for Controlled Drug Delivery

**DOI:** 10.3390/jfb13040188

**Published:** 2022-10-14

**Authors:** Shiyi Huang, Xianting Ding

**Affiliations:** State Key Laboratory of Oncogenes and Related Genes, Institute for Personalized Medicine, School of Biomedical Engineering, Shanghai Jiao Tong University, Shanghai 200030, China

**Keywords:** drug delivery, barriers, nanoparticles, microneedles, hydrogel

## Abstract

Rapid advances in nanotechnologies are driving the revolution in controlled drug delivery. However, heterogeneous barriers, such as blood circulation and cellular barriers, prevent the drug from reaching the cellular target in complex physiologic environments. In this review, we discuss the precise design of nanotechnologies to enhance the efficacy, quality, and durability of drug delivery. For drug delivery in vivo, drugs loaded in nanoplatforms target particular sites in a spatial- and temporal-dependent manner. Advances in stimuli-responsive nanoparticles and carbon-based drug delivery platforms are summarized. For transdermal drug delivery systems, specific strategies including microneedles and hydrogel lead to a sustained release efficacy. Moreover, we highlight the current limitations of clinical translation and an incentive for the future development of nanotechnology-based drug delivery.

## 1. Introduction

Mobilizing cell and microenvironment responses through drug delivery potentially constitutes a prevailing strategy to fight against disease in recent years. A precise and effective drug delivery system should improve the bioavailability of administered drugs by enhancing drug efficacy, improving cell uptake, preventing the premature degradation, maintaining drug release at target site, and reducing the side effects [1,2].

Conventional methods of drug delivery, such as chemotherapeutics, are effective to decrease lesion area burdens by triggering malignant cell death. However, conventional methods possess several drawbacks due to the administration approach such as oral administration and intravenously injection. Firstly, drugs may become less effective because the difficulty to achieve targeted cells or organ [3]. Drugs may also get inactivated or degraded while crossing several biological barriers throughout the digestive system or blood circulation. Secondly, conventional methods of drug delivery often lead to a burst release of drug instead of sustained release, which is accompanied with increasing cytotoxicity and side effects [4]. Finally, traditional methods are limited owing to drug’s poor solubility, stability and undesired release profiles [5]. Besides, conventional methods often require daily medication or injection according to the circadian behavior of the disease, e.g., in the case of insulin injections, which increases inconvenience to patients. In recent decades, the emergence of nanotechnologies and the prevalence of precision medicine has accelerated the development of drug delivery.

Recently, the burst of biopharmaceutical drugs, including peptides, monoclonal antibodies and recombinant proteins, also poses challenges because of their structural complexity and poor membrane permeation [4,6,7]. Therefore, nanotechnology-based drug delivery platforms are improved for overcoming these limitations through cell-specific targeting, molecular transport to specific organelles, and prolonged circulation times [8,9,10]. For example, nanoparticles (NPs) have shown significant promise to improve the stability and solubility of encapsulated carrier and then promote transport across cell membranes to enhance efficacy and targeting ability [3]. Moreover, nanotechnologies for drug delivery have been precisely designed to ensure the maximum potency, efficacy, and durability of therapeutic drugs to the target sites with minimum side effects [11]. Furthermore, nanotechnology is proven to be a powerful strategy for manipulating drug releasing in the desired dosage, enhancing efficacious cell response, and averting repeated drug administration or injection [12,13].

The evolution of precision medicine also promotes the advances in nanotechnology-based drug delivery. After the creation of the Precision Medicine Initiative (PMI) in 2015 [14], precision medicine put emphasis on tailoring specific treatment plan to individual by accounting for multiple genetic and epigenetic characteristics [15]. Nanotechnologies for drug delivery are designed to overcome heterogeneity among patients because of their engineering functional and structural properties, which significantly improves drug specificity, optimizes dosing, and combinatorial strategies [16,17,18].

Here, this review focuses on advances in nanotechnologies which could improve the efficacy, quality, and durability of drug delivery. We firstly focus on several biological barriers, including circulation and cellular barriers, that drug delivery needs to overcome. Furthermore, we discuss the recent progress of in vivo and transdermal implementation of drug delivery based on nanotechnologies including microneedles and hydrogels. In this review, we highlight the fact that the precise design of nanotechnologies offers significant potential in nanomedicine translation, providing a perspective on the future development of drug delivery.

## 2. Biological Barriers

Common obstacles, such as insufficient membrane transport properties and low biodistribution, are associated with the efficacy of conventional drug delivery. Hence, nanotechnology-based drug delivery platforms, especially drug-containing NPs offer better pharmacokinetic parameters. For example, the platforms improve the accumulation of drugs in the tumor site through enhanced permeability and retention (EPR) [16].

Despite their potential for improvement of accumulation at interested sites, the platforms still face a complex physiologic environment, preventing drugs from reaching their cellular target. As drugs are often delivered through various routes, including intravenous injection, transdermal administration, intramuscular injection, oral administration, and inhalation, we present dominating barriers on circulation and cellular levels that nanoplatforms need to overcome from administration to arrival at sites.

### 2.1. Cellular and Intracellular Barriers

The major obstacle of nanotherapeutic delivery is site-specific, which may cause (1) inadequate therapeutic dosage of drugs at disease sites and (2) nanotoxicity caused by accumulation in healthy organs or blood circulation. Hence, formidable challenges continue to face drug delivery. The entry of nanomedicine is designed to contact with cells of interest, activate cell endocytosis, and conduct transport into internalization through the deformation movement of the cell membrane. However, there still exist numerous cellular and intracellular barriers that influence the functional delivery.

Firstly, cell membrane acts as a barrier of drug delivery. Specifically, the phospholipid bilayers with hydrophilic heads and hydrophobic tails control selective permeability of nanomedicine, biomolecules and nano-cargoes. Therefore, understanding the fundamental mechanisms of cellular uptake is a central paradigm in nanotechnology-based drug delivery system [19]. The cellular uptake routes can be categorized into two major groups: (1) endocytosis-based entry and (2) direct cellular uptake of drug-containing NPs.

Endocytosis is a process whereby cells engulf exogenous substance by the invagination of cell membrane and budding off inside the cells [20]. Endocytosis may selectively switch to phagocytosis or pinocytosis when cells upon encountering a target substance of different size. Accordingly, nanomedicines are internalized, then transported to intracellular vesicles, including phagosomes and endosomes [21]. Besides, the interactions between nanomedicine and the mononuclear phagocyte system (MPS) determine the fate and biodistribution of administrated NPs in vivo [22]. The macrophages of MPS, which are typically resident macrophages of the liver, spleen, and lymph nodes, respond rapidly to remove NPs from the bloodstream after intravenous administration [23]. Generally, the blood half-life of ultra-small-scale NPs (<5 nm) ranged from 5 min to 4 h, while the duration time increased to 5–6 h of nanometer-scale NPs (100–200 nm) [24,25,26,27].

### 2.2. Blood Circulation and Clearance

After the NPs undergoing uptake by the MPS, blood vessel fluid dynamics may determine the fate of NPs. Specifically, both physical and biological barriers, including shear forces and protein adsorption, limit the efficiency of administrated nanomedicine that reach target site or organ [28]. In this section, we discuss the blood circulation and clearance, as well as corresponding strategies with efficiency gains.

For physical barriers, drug-containing NPs undergo varying flow rates which induce shear stress after administration [29]. The fluid forces may strip the surface coatings of drug delivery vehicles and prevent them from localizing to the target tissues [6]. Consequently, nanoplatforms or the cargoes are damaged, which leads to extravasation [30]. Therefore, physical barriers of blood circulation significantly influence the delivery and distribution.

For biological barriers, drug-containing NPs encounter various biomolecules or cells in blood (Figure 1A). Since nanomedicine enters blood as an exogenous substance, a large number of proteins in blood will bind to the surface of NPs. At the same time, immune cells, such as macrophages and neutrophils, with some specific surface receptors recognize these proteins and activate the phagocytosis function of relevant immune cells. Given the fate of nanomedicine, they are degraded by lysosome and finally cleared from blood circulation. In addition, some phagocytes have great clearance capability against the exogenous nanomedicine in blood circulation, e.g., the MPS composed of Kupffer cells in the liver and giant cells in the spleen and bone marrow, respectively. Therefore, some small-molecule exogenous substances or ultra-small-scale NPs (less than 5 nm) are quickly filtered out of blood through the glomerulus in the kidney during blood circulation, and then excreted through urine within 7 days [31].

To overcome the aforementioned barriers, the characteristics, such as size, shape and surface modifications are precisely engineered upon the specific physiochemical properties for drug delivery (Figure 1B). Firstly, the size of NPs is a controllable parameter that can be tailored for biodistribution in vivo. For instance, NPs with diameter ranging from 20 to 100 nm generally present long durability in the circulation [32]. However, they would bind with vascular endothelial cells, which often leads to nonspecific accumulation in lung, liver, and spleen [33]. Next, the shape of NPs influence hemorheological dynamics and cellular uptake. As an example, spherical NPs accumulate preferentially within the core of a blood vessel [3]. Then, the surface charge is carefully designed to improve the accumulation at target sites. NPs with negative surface charge (−10.6 mV) were reported to have long circulating half-lives of 18.8 h, while the NPs with neutral surface charge (1.3 mV) have 17.7 h [8]. Furthermore, different strategies of functionalizing NPs are also well designed. For example, to avoid rapid clearance, many NP formulations incorporate PEG, a polymer with very strong hydrophilic ability, as a coating. The binding effect of nanomedicine and serum proteins can be significantly reduced, thereby reducing the uptake of mononuclear giant cells [34,35].

Therefore, the nanomedicine that enter cells through endocytosis must be able to effectively escape from the lysosomes to avoid the degradation of drug active components and loss of function, especially the nucleic acid drugs. To reduce nanomedicine MPS sequestration, physicochemical properties of NPs, such as shape [36,37], size [38,39], surface modifications [40], hydrophobicity [41], and elasticity [42], have been precisely engineered to minimize opsonization.

## 3. Drug Delivery In Vivo

The nanotechnology-based drug delivery platforms have intrinsic issues in maximizing the drug functions. Over the past decade, researchers have applied advanced nanotechnologies to solve the problem of enzymatic degradation and cellular uptake. Besides, both chemical and biopharmaceutical drugs still have poor stability as well as solubility and toxicity problems. Therefore, an ideal drug delivery system would be capable of not only preserving the properties of drugs, but also penetrating target cells and controlling drug release [43]. In this section, we discuss different strategies of drug delivery in spatial-, temporal-, and dosage-controlled routes.

### 3.1. Stimuli-Responsive Nanoparticles

Nanoparticles (NPs), due to their smaller scale, can improve the drug delivery performances and overcome barriers such as intracellular trafficking. Considering the drug release requirements, optimal NPs will fulfill the following conditions: (1) high payload and effective delivery, (2) excellent targeted ability or highly selective manner, (3) programmable surface for improving in vivo stability, (4) lower toxicity with progressive biodegradation and high possibility to excrete, and (5) significant durability for cell response without burst effect [4,44,45]. Stimuli-responsive NPs are sensitive to specific endogenous stimuli. Thus, the leakage of drugs only occurs under particular condition, which is a guarantee to the potency, efficacy, and durability of the drug delivery system.

#### 3.1.1. PH-Sensitive Responsive Nanoparticles

Taking advantage of various pH gradients in pathological sites, including organs (such as stomach, pH = 1.0–3.0; duodenum, pH = 4.8–8.2; colon, pH = 7.0–7.5) and intracellular compartments (such as lysosomes, pH = 4.5–5.0; endosomes, pH = 5.0–6.5), pH-responsive drug delivery systems are appealing to control the release of drug to targeted places [46]. The condensed tumor extracellular matrix (ECM, pH = 6.5–7.2) is considered to be a key factor associated with the irregular tumor microenvironment [47]. Hence, pH-sensitive NPs are designed towards the ECM since it features reduced pH values. The ECM is a hydrated gel-like matrix, which consists of a crosslinked network of collagen, hyaluronic acid (HA), elastin fiber, glycoprotein, and proteoglycans [48,49]. The main strategy against ECM focuses on the degradation of ECM so as to increase the penetration of drugs and modulate the hypoxic tumor microenvironment [50,51]. Wang and co-workers [52] conjugated dextran (DEX) with hyaluronidase (HAase) through a pH-sensitive traceless linker, fabricating the DEX-HAase NPs (Figure 2A). Their NPs showed blocked enzyme activity under physiological pH and released native free HAase under weak acidic pH within tumor microenvironment.

In addition, bromelain (Br) was crosslinked and then encapsulated doxorubicin (DOX) into NPs, which successfully degraded ECM, entered into nucleus by passive diffusion, and released DOX in tumor region (Figure 2B) [53]. Besides, the co-delivery of multiple drugs by NPs is adopted in therapy. A pH-sensitive mesoporous silica NP (MSN-CS-LA) controlled-release system for co-delivery of ursolic acid and sorafenib was developed for targeting asialoglycoprotein receptor over-expressing hepatocellar carcinoma cells (Figure 2C) [54].

#### 3.1.2. Thermosensitive Drug Delivery System

Thermosensitive drug delivery has received increasing academic interest in oncology, providing the potential of combination between chemotherapy and thermotherapy. Generally, thermosensitive NPs should have the capability of retaining the load at body temperature (37 °C) and releasing the drugs rapidly within heated tumor sites (40–42 °C).

Generally, thermo-responsiveness is mainly governed by a nonlinear sharp temperature change between target sites and their surroundings. When triggered with near-infrared (NIR, λ = 700–1100 nm) light, thermosensitive NPs possess rapid and quantitative drug release performance [55,56]. To integrate real-time monitoring and therapeutic functions into a single nanoagent, bismuth selenide (Bi_2_Se_3_) NPs coated with polydopamine (PDA), human serum albumin, and DOX were applied simultaneously for photothermal therapy (PTT), chemotherapy, and dual-modal imaging (Figure 3A) [57]. Additionally, combating bacterial infection and promoting wound healing are also promising applications. An injectable hydrogel composed by mixing ciprofloxacin (Cip, a potent antibiotic)-loaded NPs and glycol chitosan realizes drug delivery and hyperthermia-assisted bacterial inactivation (Figure 3B) [58].

Recently, the mitochondrial temperature-dependent delivery has aroused interest because the mitochondrial function is closely associated with carcinogenesis. The temperature of mitochondria has been reported higher than ambient temperature and approximately reaches 48 °C [59,60]. In addition, the temperature of mitochondrial in cancer cells is also higher than that in normal cells [61,62]. Taking advantage of the feature, a thermos-responsive nanocarrier that loaded DOX was designed to achieve mitochondria in DOX-resistant tumor sites recently (Figure 3C) [63].

To summarize, we discuss that thermosensitive drug delivery significantly enhanced pharmaceutic effects in both organs and organelles. Otherwise, the challenge of thermosensitive NPs still lies in the design of nanoagents which combine sensitivity and safety for clinical applications.

#### 3.1.3. Photo-Triggered Response

Owing to the advantage of non-invasiveness, light activation can readily control the drug release with high spatiotemporal precision [64]. Light-triggered drug delivery responds to illumination of different wavelength including ultraviolet (UV), visible and NIR light [65]. Therefore, the light-triggered response may offer some advantages: (1) optional treatment frequency on either one-time or repeatable therapy; (2) controllable time of drug release; (3) low toxicity of drug carriers when circulate in the body.

For ocular diseases, light activation has been demonstrated to be a less invasive manner to bypass the blood–retinal barrier (BRB), which is also considered to be more patient-friendly. Currently, red light [66], green light [67], blue light [68], and UV [69,70] irradiation provide significant potentials with low penetrating ability. As shown in Figure 4A, the clathrin-like triangular molecules could self-assemble with hydrophobic drugs to form stable NPs. Then, NPs loaded with DOX rapidly disintegrated, released, and accumulated at retinoblastoma sites after intravenous injection with the stimulation of green light [67].

Moreover, light activation provides an optimal choice for the sequential delivery system. Specifically, one drug is released firstly to receive the extracellular targets of cells. Then, the other drug, along with the nanocarrier, is internalized into cells for the intracellular target treatment [73]. As shown in Figure 4C, Lai and co-workers [71] proposed a sequential delivery strategy which mainly relies on a light-responsive liposomes encapsulated with sunitinib [74] (antiangiogenic inhibitor for multiple cancers).

Furthermore, the photo-triggered drug delivery system also combines with other chemotherapy or strategy. For instance, the double-helical microswimmers in the form of a magnetic chitosan nanocomposite could release drugs under the trigger of near-infrared light, along with being actuated and controlled under the special-manipulated magnetic field to target site (Figure 3B) [72]. In another case, mesoporous silica nanoparticles encapsulated pharm molecules under glutathione- and light-controlled dual activation [75].

#### 3.1.4. Enzyme-Sensitive Drug Delivery

Enzyme-sensitive drug delivery represents a promising strategy to gain control over the formation or disruption of nanostructures owing to its substrate specificity and high selectivity [76,77]. Enzymes, such as proteases, esterases, or glycosidases, are often exploited as a tool for targeted therapy due to their concentration gradients between tumor cells and normal cells [78,79].

There are two major approaches of enzyme-responsive drug delivery. On the one hand, the NPs are fabricated with materials sensitive to enzymatic transformation. Thus, NPs would release the drugs upon encountering the specific enzyme. In this case, polymer-based NPs [80], liposomes [81], mesoporous silica NPs [82], and semiconducting NPs [83] (such as CdS and CdSe) are usually adopted. For instance, the core-shell NPs which are composed with polymer core and hyaluronic shell layer would release drugs after the degradation by hyaluronidase (HAase) (Figure 5A) [84]. Similarly, liposome encapsulating collagenase NPs has the ability to disassemble the dense pancreatic ductal adenocarcinoma (PDAC) collagen stroma and increase drug penetration into tumor sites (Figure 5B) [81]. On the other hand, the surface of NPs is carefully modified with molecules that produce changes (mainly are physical properties) under particular conditions [78,85,86].

To sum up, although enzyme responsive NPs systems have been attempted for therapy, shortcomings, such as a lack of targeting specificity and requirement of long reaction time, limit the clinical applications.

### 3.2. Carbon-Based Platforms of Drug Delivery

Over the past decade, nanotechnologies based on carbon materials, including carbon nanotube (CNT) [87], graphene and its derivatives [88], and carbon dots (CDs) [89], have aroused in the biomedical application. The surface area of carbon materials possesses high potential of functionalization and modification, which may enhance the solubility, stability, and biocompatibility in physiological environment. Therefore, a significant number explorations on therapeutic agent delivery systems by means of carbon materials have been carried out [90]. In this section, we discuss recent advances in carbon-based structures for drug delivery.

Graphene oxide (GO) is a typical two-dimensional nanomaterial. Many studies have demonstrated that GO of dimensional size between 20 and 300 nm is readily perceived by the immune system [91,92]. The edges of GO are an alternative location easily employed to link a wide range of biomolecules and moieties [93,94]. Lazaro and co-workers [95] investigated whether GO flakes could distribute deeply and homogenously throughout glioblastoma tumors and subsequently target the myeloid compartment (Figure 6A). They studied the interactions of GO flakes with in vitro and in vivo models of glioblastoma. In vitro, GO flakes translocated deeply into the spheroids with little internalization into tumor cells. In vivo, GO showed extensive distribution throughout the tumor and also presented low impact on tumor growth and progression. Li and co-workers [96] reported a new type of antineoplastic agents, using GO loaded with photosensitizer and DOX as a core and the red blood cell membrane as a shell (Figure 6B). In addition, the antineoplastic agent is decorated with folic acid for the selective recognition of tumor cells via a lipid-insertion approach.

Reduced GO (rGO) is obtained from GO, however, rGO exhibits different physicochemical properties from GO such as lower degree of oxidation and higher electrical conductivity [101]. Recently, it was reported that rGO was associated with reactive oxygen species (ROS) [97,98]. rGO was demonstrated that it could serve as a neurotransmission modulator and rGO could be oxidized via cellular ROS (Figure 6C) [97]. Oxidized rGO was also proven to exert depressant effects on neurotransmission because of the blockage of synaptic vesicle docking and fusion induced through a disturbance of actin dynamics. Additionally, as shown in Figure 6D, PEGylated rGO nanosheet (rGO-PEG) was designed as a multifunctional nanovaccine platform for direct delivery of neoantigens and adjuvants to lymph nodes (LNs) [98]. Moreover, rGO-PEG induces intracellular ROS in dendritic cells, guiding antigen processing and neoantigen-specific T cell response.

Carbon nanotube porins (CNTPs) are promising CNT-based materials for drug delivery since they are short pieces of CNTs inserted into lipid membranes [102,103]. CNTPs could facilitate the fusion of lipid membranes, thus enhancing cellular uptake and the biocompatibility of CNTs [104]. Ho and co-workers demonstrated that liposomes studded with 0.8-nm-wide CNTPs were effective in delivering the drug to cancer cells and killing tumor cells (up to 90%) from fusion kinetics data and coarse-grained molecular dynamics (Figure 6E) [99].

Carbon dots (CDs) are emerging nanomaterials, showing high drug loading capacity. The extreme small size of CDs (<20 nm) enables good ability to overcome the biological barrier, but it also leads to rapid elimination from the circulation. As shown in Figure 6F, CDs decorated with PEG and aminoethyl anisamide (AEAA) are crosslinked by peptides to produce the mesoporous nanostructures. Next, DOX and Fe ions, a trigger boosting ROS intracellularly, are immobilized on the surface of mesoporous nanostructures, while LOS (an inhibitor mitigating tumor hypoxia) are encapsulated in the mesopores [100].

In conclusion, carbon-based therapeutics can provide a significant breakthrough in the field of drug delivery. Nevertheless, none of the carbon nanomaterials has been evaluated or applied in a clinical trial. Therefore, more efforts are still required on unresolved critical issues to motivate the clinical translation of carbon-based therapeutics.

## 4. Transdermal Drug Delivery System

Transdermal drug delivery system (TDDS) has been proven to be a patient-friendly administration route. The advantages of TDDS can be summarized in four major points: (1) non-invasive and painless administration; (2) can be self-administered; (3) avoids hepatic first-pass metabolism; (4) sustained release with single administration [105,106]. To overcome the barrier of skin, TDDS needs to cross three layers: epidermis, dermis, and hypodermis [107]. In this section, we discuss two typical TDDS routes: microneedles and hydrogels.

### 4.1. Microneedles

Microneedles (MNs) are the most widely used nanotechnology for TDDS. To date, MNs are often designed to align as an array on a patch, containing drug or biologics (insulin, vaccine, antibodies and peptides) [108]. Micron-sized needles penetrate the cutaneous stratum corneum and do not touch the nociceptive nerves, thus avoiding the pain of needling, which deliver drugs to local or systemic effects within minutes after being applied to the skin. At the same time, the average diameter of the micropores left in the skin barrier after MNs extraction is about half that of the needle due to the physiological elasticity of the skin [109]. In this section, we discuss several MNs and their recent advances with illustrations.

#### 4.1.1. Solid MNs

Solid MNs deliver the drug with passive diffusion to skin layers, and part of the needles is usually dissolvable when penetrating in the cutaneous stratum corneum [110]. For instance, a single removable transdermal patch bearing MNs loaded with insulin and a non-degradable glucose-responsive polymeric matrix was reported recently (Figure 7A) [111]. This MNs could regulate blood glucose in insulin-deficient diabetic in both mice and minipigs model. Interestingly, MN arrays are designed to be combined with a portable smartphone to precisely control transdermal drug delivery [112]. In this study, the MN array patch was pressed on the skin to create microholes, followed by the iontophoresis delivery of insulin in an electrical-controlled manner (Figure 7B). Besides, Bilal et al. [113] found gelatin methacrylate crosslinked with polyethylene glycol diacrylate is potent for fabricating MN arrays. In this study, the incorporation of MoS_2_ nanosheets (as a photothermal component) into hydrogels results in on-demand release properties.

Moreover, MNs are also promising delivery system that combine with nonphysical modes of drug delivery enhancement. In this study, Chen et al. [114] reported a novel dynamic omnidirectional mucoadhesive MNs system capable of prolonged gastric mucosa fixation which is inspired by the thorny-headed intestinal worm. Additionally, this MNs system can also be applied as a targeted capsule system to generate and maintain a privileged region in the gastrointestinal tract.

#### 4.1.2. Porous MNs

Porous MNs have porous tips through which liquid or dry drugs diffuse [66]. Silicon is a widely used material for porous MNs. To improve the biocompatibility of silicon, Kim and co-workers [115] designed a bioresorbable, miniaturized porous-silicon MNs, and it could completely dissolute with saline solution within 1 min (Figure 7C). The MNs are proven to continuously release the preloaded drug cargos over days at a controlled rate in a murine melanoma model.

Polymer-based porous MNs have also been used to deliver drugs due to the varying degrees of porosity. The limitation mainly lies in the complexity of manufacturing technology able to create core-shell or reservoir-based structures [120]. Tran et al. [116] reported a transdermal core–shell microneedles, which were fabricated by the micromoulding and alignment of Prevnar-13 (vaccine against the bacterium Streptococcus pneumoniae) cores and shells made from poly(lactic-co-glycolic acid) (Figure 7D). They also verified that the immune responses of MNs were similar to that after multiple subcutaneous injections over a period of a few days to more than a month from merely a single administration. Besides, Kusama et al. [117] reported an ion-conductive porous MNs containing interconnected micropores for improving iontophoresis. Iontophoretic TDDS is a new approach to accelerate transdermal penetration by a continuous low voltage current, which has also been commercialized successfully (Figure 7E) [121,122,123]. In this research, their MNs brought three innovative advantages: (1) lowering the transdermal resistance by low-invasive puncture in stratum corneum; (2) transporting of larger molecules through the interconnected micropores; (3) generating electroosmotic flow, thus improving the transdermal molecular penetration.

#### 4.1.3. Coated MNs

In order to realize the possibility of deliver multi agents in a single patch, MNs are fabricated with drug dispersion layer to form coated MNs [124]. Therefore, the drug from the coated layer may dissolute rapidly. For application in vaccine, Duong and co-workers [125] designed a smart DNA vaccine delivery system in which DNA vaccine was laden on MNs. In this case, their MNs were assembled with layer-by-layer coating of a pH-responsive polymer and vaccine adjuvant (poly (I:C), a toll-like receptor 3 agonist). For application in rapid hemostasis, Zhang and co-workers [118] reported a dodecyl-modified chitosan (DCS)-coated multilayer MNs patch inspired by the architecture structure of pagoda (Figure 7F). This MNs patch was fabricated by a step-by-step mold replication, then MNs were coated with DCS. Researchers evaluated coated MNs on the tissues in rabbit model, indicating great fixation and hemostasis capability. Besides, Perry and co-workers [119] fabricated a faceted MNs by three-dimensional (3D)-printing technique (Figure 7G). Compared with a smooth square pyramidal structure, the surface area increased more than 21.3%, which led to a higher drug payload.

### 4.2. Hydrogels in Transdermal Drug Delivery System

Hydrogels are appealing types for drug delivery systems due to their high-water contents (70–99%) [126]. Hydrogels provide physical similarity, and the intrinsic porous and hydrophilic structure of hydrogels guarantees gas exchange and fluid balance on the skin [127,128]. According to the bind of polymer network, the hydrogels can be divided into two categories: covalent hydrogel and non-covalent hydrogels. Covalent bonds (e.g., carbon-carbon bond and amide bond) link monomer units into the polymer chain, then crosslink other similar polymer chains to constitute a polymer network, thus forming the covalent hydrogel. By contrast, non-covalent bonds such as amine and hydroxyl can interact strongly with other aggregates of non-covalent bonds to form a strong polymer complex. Generally, the bond energy of a covalent bond is 80–320 kT, while that of a hydrogen bond is only 1–50 kT [129,130]. Interestingly, multiple non-covalent interactions act as highly stable [131].

Moreover, transparency property is also an interesting aspect to monitor the delivery results [132]. Hence, all these advantages make hydrogels attractive material for many therapeutics. In this section, we discuss recent advances made regarding covalent hydrogels and non-covalent hydrogels for TDDS.

To achieve different mechanical properties and drug release behavior of hydrogel, the degree of crosslinking could be tuned. Luo et al. [133] reported a polymer patch for sustained drug delivery based on Gelatin methacryloyl (GelMA), a material that could be crosslinked by ultraviolet (UV) or visible light in the presence of photoinitiators. Besides, Jung and co-workers [134] reported a tissue adhesive hydrogel, which consisted of polyacrylamide/polydopamine (PAM/PDA) hydrogels embedded with extra-large pore mesoporous silica nanoparticles (Figure 8A). Notably, PAM hydrogels feature mechanical and chemical advantages: (1) high degree of recovery after stress tests; (2) controllable rigidity of gel [135,136]. In this study, the optimized condition enables a 4.9-fold increase in adhesion energy of PAM/PDA hydrogels compared to the control group.

Furthermore, covalent hydrogels are combined with iontophoretic TDDS to conduct a wearable device for closed-loop motion detection and therapy (Figure 8B) [137]. In this study, the hydrogel-based soft patch with side-by-side electrodes was designed to enable noninvasive iontophoretic TDDS, along with the motion sensor and energy harvester that can convert biomechanical motions into electricity. Similarly, An and co-workers [138] developed an iontophoretic TTDS consisting of an electroconductive hydrogel (polypyrrole-incorporated poly(vinyl alcohol), PYP) and a portable battery (Figure 8C).

In general, although hydrogel based TDDS have been translated in the clinic, there are still some challenges existing in terms of fabrication process, storage, and higher cost [126,139]. Meanwhile, the release of multiple drugs at different rates in hydrogels remains a challenge.

## 5. Conclusions and Future Prospects

The review has discussed numerous nanotechnology-based drug delivery systems from the barriers to recent advances. Barriers, including blood circulation and cellular uptake, are complex to overcome because both stages of disease progression and patient heterogeneity may lead to different physiologies. Assessing the actual patient status and utilizing precision medicine therapeutics remain critical [140]. Moreover, we discuss the characteristics of NPs, such as size, shape, and modification, are determinable for the delivery targeting (both organs and cells) and efficacy. Furthermore, microneedles and hydrogels allow flexibility in transdermal drug delivery system.

However, we researchers need to soberly be aware that the translation of nanotechnology-based drug-delivery systems from the bench to the bedside is not straightforward at present. Our final proposal is to develop the strategies of nanotechnologies for controlled drug delivery, to enhance the efficacy, quality, and durability, so as to reach the target cells or organs. For example, NPs may trigger subtle effects, such as cascades of cell signals, and long-term effects on gene expression. Therefore, understanding the interaction between nanomaterials and the human body may facilitate the translation of drug delivery for clinical application.

Remarkably, tuning the physicochemical properties of NPs is of great significance in improving the therapeutic effectiveness of drug delivery systems and avoiding immunotoxicity effects. For example, copper oxide nanoparticles (CONPs) are a kind of representative azurite ore, as a stable nanodrug in various solutions [141]. CONPs can target mitochondria, induce ROS production, and ultimately lead to cell apoptosis and death. Therefore, differences in the amount of ROS in the cellular environment can lead to the cytotoxicity, DNA and protein damage, and lipid peroxidation regulation of some cellular events, although the presence of ROS is important for some metabolic functions in the body. Besides, the dissolution of NPs sometimes also induces cytotoxicity [142]. For instance, copper toxicity can be inhibited by gene disruption of the enzyme required for lipoic acid synthesis or by the individual lipoacylase itself [143]. Furthermore, approaches, such as meta-analysis or data-mining, can reveal potential cytotoxicity of NPs. Labouta et al. [144] showed that material type ranked the most important parameter which determined the cytotoxicity, along with diameter and exposure time.

Several issues still remain to be overcome for successful clinical translation. For drug delivery in vivo, the stability and specific responsive property of nanoplatforms may limit the drug loading efficacy. Therefore, the pharmacological activity actually released at target organs or cells is determined with a successful formulation strategy. For transdermal drug delivery systems, the advantages of specific strategies, including microneedles and hydrogel, lead to sustained release efficacy and patient compliance. Although the safety of transdermal drug delivery system, researchers still aim to improve the efficient delivery of small molecules through oral or parenteral routes [145].

Furthermore, the studies among protein corona aroused discussions regarding the enhancement of cellular uptake over the decades. Protein corona based NPs are verified to be easily adsorbed biomolecules, especially proteins, forming protein coronas in the circulation system [146]. These coronas serve a dual purpose: (1) they isolate NPs from organisms; (2) these biomolecules are important mediators of certain life processes, activating disordered biological functions [147]. Tenzer et al. [148] found that plasma exposure time significantly affected the abundance of factors associated with processes, such as complement activation, cell death, immune response, coagulation, and acute phase response. The subsequent experiments showed that the rapid formation of coronas could affect hemolysis, platelet activation, nanoparticle uptake, and endothelial cell death.

As we have discussed in this review, the therapeutic efficacy and durability based on nanotechnology can be improved by optimizing this specificity and local activity of NPs delivery systems, enhancing the fabrication process of MNs and tuning the agents and doses of hydrogels. Precise design will ultimately generate the translation of nanotechnologies into clinical.

## Figures and Tables

**Figure 1 jfb-13-00188-f001:**
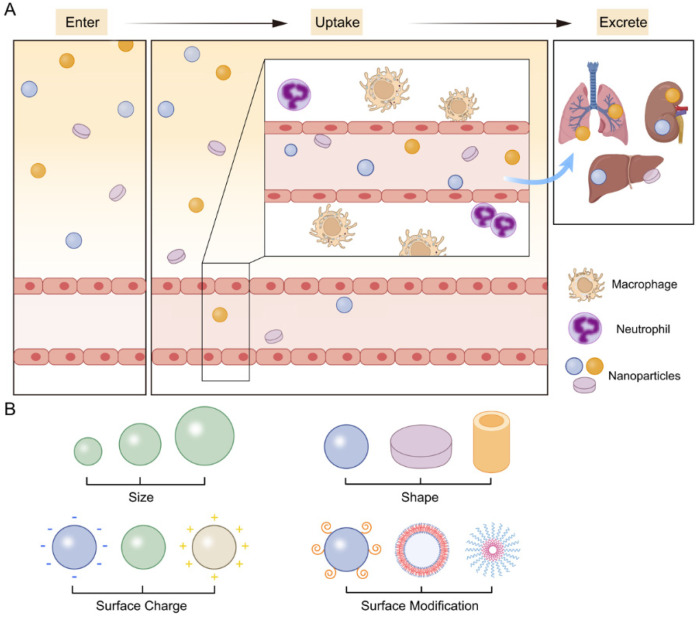
(**A**) The biological barriers of drug-containing NPs of blood circulation. Upon intravenous administration, NPs enter blood vessels and undergo opsonization according to their own physicochemical characterization. Subsequently, the resident macrophages of the MPS uptake NPs. (**B**) The physicochemical properties of NPs including size, shape, surface charge, surface modification are key factors of the cell uptake.

**Figure 2 jfb-13-00188-f002:**
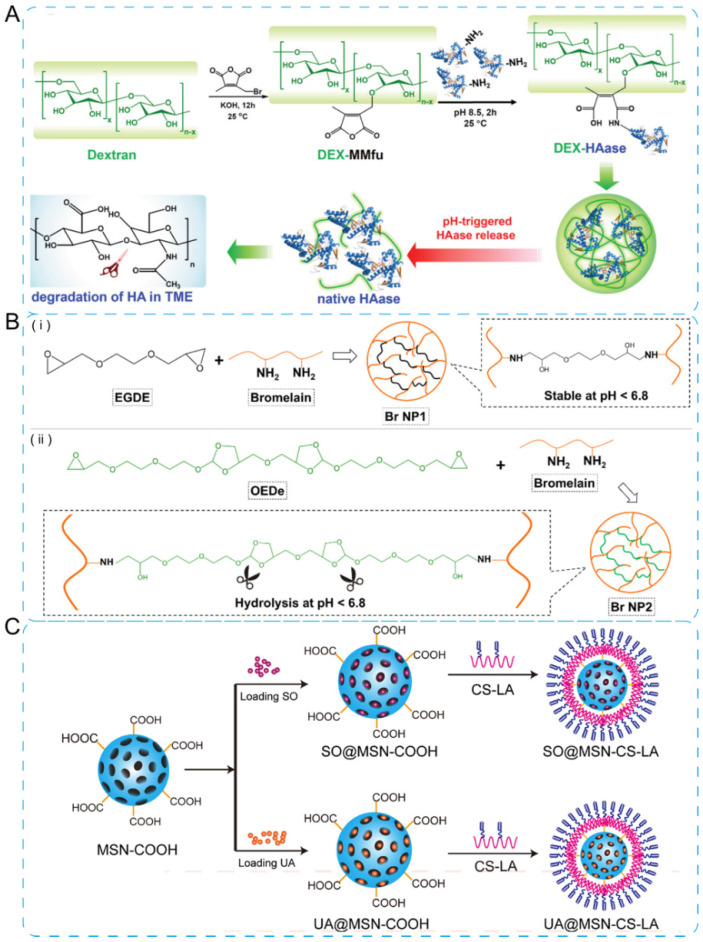
pH-sensitive NPs. (**A**) The mechanism of DEX-HAase fabrication and pH-triggered drug release. Reprinted with permission from Ref. [52]. Copyright 2019, Wiley-VCH. (**B**) pH-sensitive bromelain nanoparticles by ortho ester crosslinkage for enhanced DOX penetration in solid tumor, with the synthetic route of non-sensitive core (i) and pH-sensitive shell (ii). Reprinted with permission from Ref. [53]. Copyright 2020, Elsevier. (**C**) pH-responsive chitosan-conjugated mesoporous silica nanocomplex for co-delivery of ursolic acid and sorafenib. Reprinted with permission from Ref. [54]. Copyright 2017, Elsevier.

**Figure 3 jfb-13-00188-f003:**
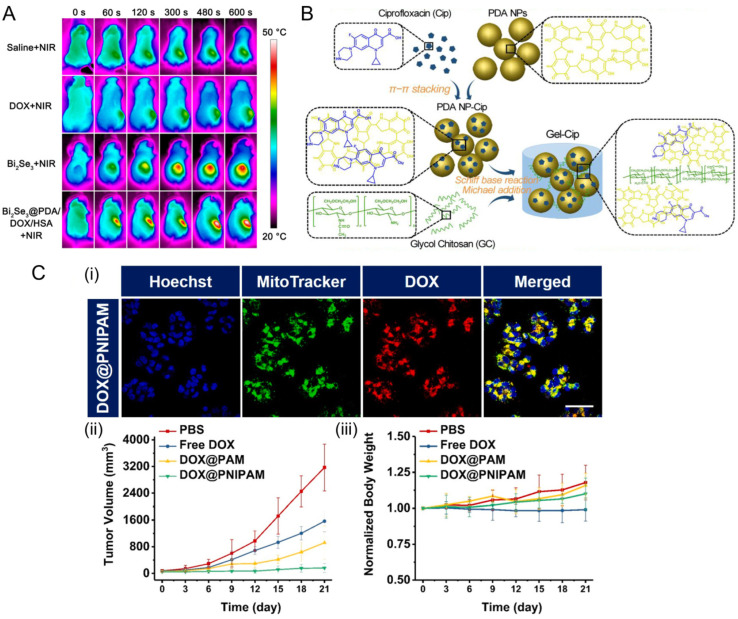
Thermosensitive drug delivery system. (**A**) Bi_2_Se_3_ NPs coated with PDA/human serum albumin/DOX inhibited tumor on HeLa tumor-bearing mice. Reprinted with permission from Ref. [57]. Copyright 2016, American Chemical Society. (**B**) The synthetic route of anti-bacteria NPs and NIR light irradiation-triggered drug release to combat bacterial infection. Reprinted with permission from Ref. [58]. Copyright 2019, Elsevier. (**C**) Thermo-responsive NPs enhance drug accumulation in mitochondria in response to mitochondrial temperature in lung cancer. In this case, poly (N-isopropylacrylamide) (PNIPAM) was used as thermosensitive nanocarrier, then DOX was loaded in to PNIPAM NPs by solvent replacement method to obtain DOX@PNIPAM (i) confocal images of H69AR cells incubated with 10 μg/mL of DOX-loaded NPs for 12 h. Scale bars: 50 μm. (ii) tumor growth profiles of mice treated with PBS, DOX and NPs. (iii) the body weight of mice after treatment. Blue, cell nucleus; green, mitochondria; red, DOX. Reprinted with permission from Ref. [63]. Copyright 2022, Elsevier.

**Figure 4 jfb-13-00188-f004:**
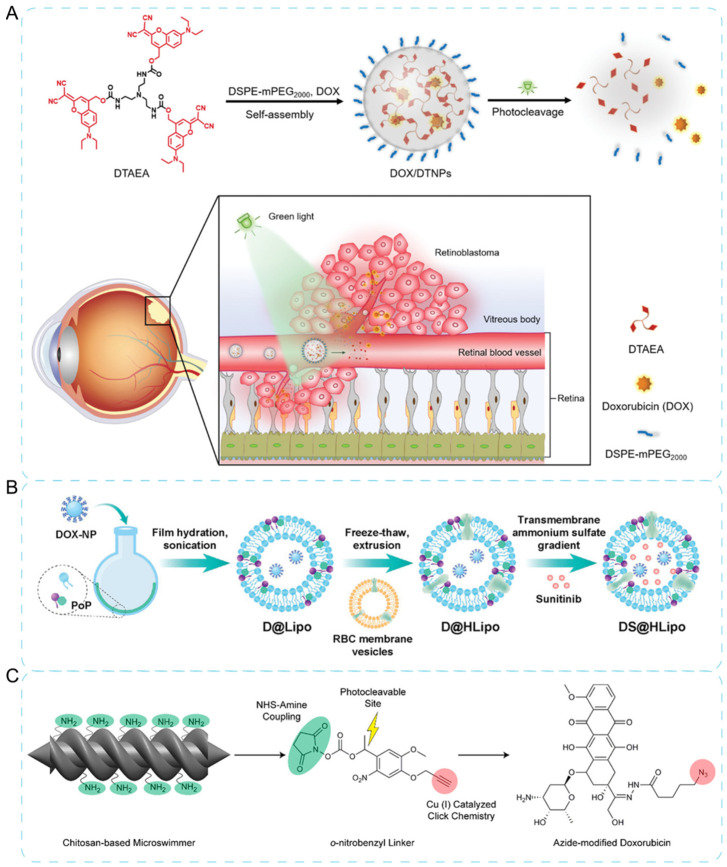
Photo-triggered therapy by NPs. (**A**) Green light-triggered chemotherapy of retinoblastoma and its mechanism of photo-enhanced drug accumulation in the eye. Reprinted with permission from Ref. [67]. Copyright 2021, Wiley-VCH. (**B**) Schematics of light-triggered sequential drug delivery based on liposomes and sunitinib. Reprinted with permission from Ref. [71]. Copyright 2022, Wiley-VCH. (**C**) The mechanism of microswimmers binding to drug. Firstly, amino groups on the nanocarrier reacted with NHS group of photocleavable linker. Next, azide-modified DOX reacted with alkyne ends of the conjugated microswimmers. Reprinted with permission from Ref. [72]. Copyright 2018, American Chemical Society.

**Figure 5 jfb-13-00188-f005:**
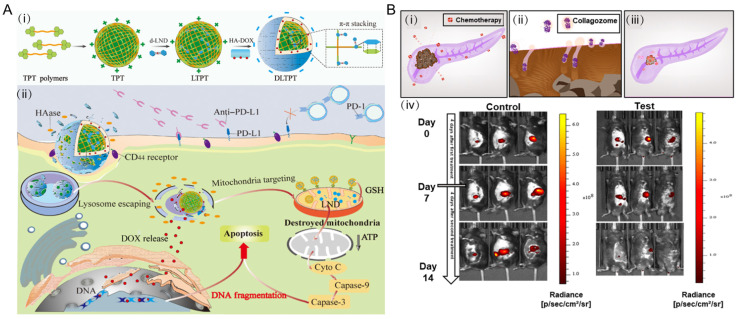
Enzyme-sensitive NPs for drug delivery. (**A**) Schematic for the self-assembly process of the enzyme-sensitive hierarchical nanoplatform. The surface layer of core-shell nanoparticles was composed of hyaluronic acid (HA). After the degradation of core-shell nanoparticles, the inner positively charged triphenyl-phosphonium derivatives particles (LTPT) were released at the first place, thus penetrating deeper into the tumor tissues. Meanwhile, the LTPT could target the mitochondria to load drugs. (i) schematic for the self-assembly process of the cascade-targeting enzyme-sensitive hierarchical nanoplatform. (ii) schematic of the combinational effects for activating the immune system to maximize the chemo-immune therapeutics. Cyto C, cytochrome. Reprinted with permission from Ref. [84]. Copyright 2021, AAAS. (**B**) Schematic of collagenase NPs for PDAC tumors and their therapeutic efficacy in mice model. (i) schematic of tumor drug resistance by ECM. (ii) proteolytic enzymes housed within nanoparticles for disassembling the collagen component of the tumor ECM. (iii) pretreatment increased tumor drug uptake of collagenase-encapsulated liposomes and collagozomes. (iv) tumor radiance decreased with images of whole-animal imaging system. Reprinted with permission from Ref. [81]. Copyright 2019, American Chemical Society.

**Figure 6 jfb-13-00188-f006:**
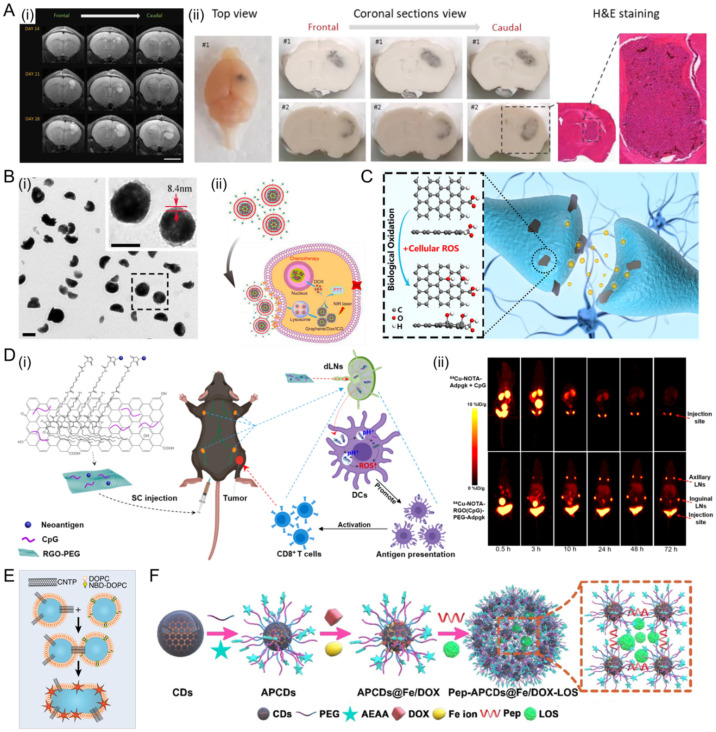
Carbon material platforms for drug delivery. (**A**) GO flakes administered intratumorally distributes extensively in a U-87 MG orthotopic xenograft. MRI assessment, photos of frozen brains (i) and H&E staining (ii) of the tissue indicate that GO flakes showed low effect on tumor growth. Scale bars: 4 mm. Reprinted with permission from Ref. [95]. Reproduced with permission. Copyright 2020, Wiley-VCH. (**B**) TEM images (i) and schematic (ii) of the antineoplastic agents in a core-shell structure. Reprinted with permission from Ref. [96]. Reproduced with permission. Copyright 2019, Elsevier. (**C**) rGO served as a neurotransmission modulator. Reprinted with permission from Ref. [97]. Reproduced with permission. Copyright 2020, American Chemical Society. (**D**) Schematic illustration of rGO-PEG nanoplatform for LN-targeted delivery of antigens and adjuvants, leading to efficient delivery to APCs in local LNs, generation of intracellular ROS, and elicitation of robust antitumor T cell immunity. (i) schematic of RGO(CpG)-PEG-neoantigen for LN-targeted delivery of antigens and adjuvants. (ii) serial PET images of C57BL/6 mice after antigen delivery. Reprinted with permission from Ref. [98]. Reproduced with permission. Copyright 2020, American Chemical Society. (**E**) Schematic of CNTPs-mediated vesicle fusion for cancer cells. Reprinted with permission from Ref. [99]. Reproduced with permission. Copyright 2021, National Academy of Sciences. (**F**) Schematic of drugs-loaded nanoassemblies of CDs. Reprinted with permission from Ref. [100]. Reproduced with permission. Copyright 2020, Wiley-VCH.

**Figure 7 jfb-13-00188-f007:**
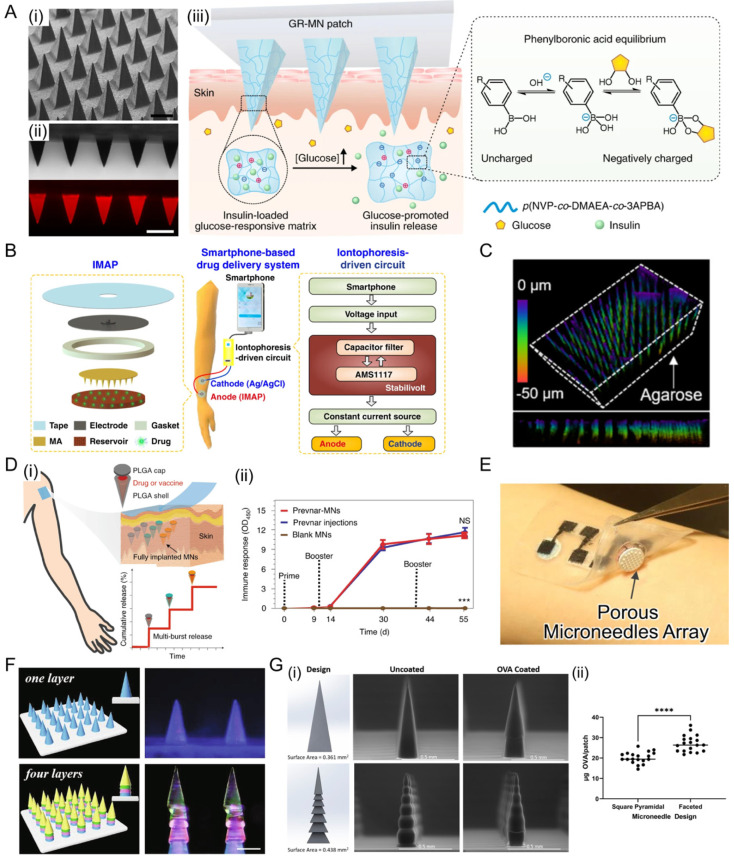
Solid, porous and coated MNs for transdermal delivery. (**A**) (i) Photograph of the GR-MN patch. (ii) scanning electron microscopy image of the microneedle array. Scale bar: 500 μm. (iii) mechanism of the glucose-triggered insulin MNs. Upon exposure to a hyperglycaemic state, the increased negative charges weakened the electrostatic interaction between negatively charged insulin and polymers, leading to the rapid release of insulin. Reprinted with permission from Ref. [111]. Copyright 2020, Springer Nature. (**B**) Schematic illustration of a smartphone-powered MNs array. It mainly consists of the MNs array, an iontophoresis-driven circuit and a smartphone. To control drug delivery, the circuit can stabilize the input voltage and output a constant current for the iontophoresis under the power supply of the smartphone. Reprinted with permission from Ref. [112]. Copyright 2020, Springer Nature. (**C**) Confocal microscopy image of the silicon MNs embedded inside an agarose gel (2.8% *w/v*). Reprinted with permission from Ref. [115]. Copyright 2020, American Chemical Society. (**D**) Schematic of the core–shell MNs (i) and the immune response accordingly in rat model (ii). Reprinted with permission from Ref. [116]. Reproduced with permission. Copyright 2021, Springer Nature. (**E**) Photo of porous MNs for enhancing iontophoresis. The white disks indicated by arrow were porous MNs array, and the black ones are biobatteries. Reprinted with permission from Ref. [117]. Copyright 2021, Springer Nature. (**F**) Schematic illustrations and fluorescence images of multilayer coated MNs patch. Scale bars: 350 μm. Reprinted with permission from Ref. [118]. Copyright 2021, Elsevier. (**G**) Design and scanning electron microscope (SEM) images of faceted MNs (i) and their accordingly increased drug payload (ii). Reprinted with permission from Ref. [119]. Copyright 2021, National Academy of Sciences. NS, no significance, *** *p* < 0.001, **** *p* < 0.0001.

**Figure 8 jfb-13-00188-f008:**
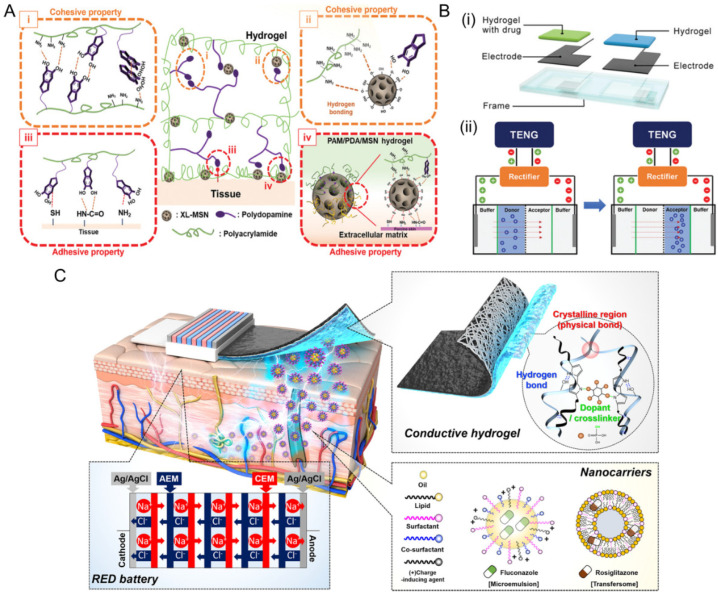
Covalent and non-covalent hydrogels. (**A**) Schematic of PAM/PDA hydrogel patch with cohesive and adhesive properties on skin tissue associated with enhanced (i,ii) cohesive property and (iii,iv) adhesive property. Reprinted with permission from Ref. [134]. Copyright 2020, Wiley-VCH. (**B**) Schematic of a wearable device, consisting of hydrogel-based soft patch and triboelectric nanogenerator. (i) schematic of the device structure. (ii) schematic of experimental setup using a diffusion cell. Reprinted with permission from Ref. [137]. Copyright 2020, Wiley-VCH. (**C**) Schematic of the RED-driven iontophoretic patch and each component. Reprinted with permission from Ref. [138]. Copyright 2020, American Chemical Society.

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
