# Peer review of "Precise Design Strategies of Nanotechnologies for Controlled Drug Delivery"

_jfb, 2022, doi:10.3390/jfb13040188_

Round 1
Reviewer 1 Report
The mini review discusses the design of nanotechnologies to enhance the efficacy, quality and durability of drug delivery. The authors discuss drug delivery in vivo, drug loaded in nanoplatforms target particular sites in a spatial- and temporal-dependent manner. They also discussed transdermal drug delivery system, specific strategies leading to a sustained release efficacy and highlighted limitations of clinical translation and introduced an incentive to the future development of nanotechnology-based drug delivery.
Author Response
We appreciate the Reviewer’s time for in-depth reading and commenting on our manuscript. We also appreciate theReviewer’s positive response. Also, we have also noticed that the Reviewer’s comment that English language and style are fine/minor spell check required, hence we have revised the manuscript and added more detailed description according to all Reviewers’ comments which indeed facilitate the readability and clearance. We sincerely hope these additional elaborations may facilitate to better address the Reviewer’s concerns.
Reviewer 2 Report
The presented articles summarized different types of nanoparticles in drug delivery. In my opinion, it is a comprehensive, and up-to-date review of relevant literature. The authors globally illustrate the application of various nanoparticles with detailed examples. However, there are several points need to be revised. Here listed my comments:
1. This is an interesting manuscript, but on some points is a bit too general. It simply summarized the different types of nanoparticles. I would suggest the authors to focus on solving detail challenges in this area instead of including all the subtypes of nanoparticles.
2. The mononuclear phagocyte system (MPS) is the newer term to represent reticuloendothelial system (RES). They described the same mechanism.
3. The two barriers discussed in this article are not very separated. The blood clearance is mediated by cellular endocytosis/phagocytosis. I strongly recommend the authors to reorganize section 2.
4. The readers will have a more comprehensive understanding of design of nanoparticles from a discussion regarding off-target cytotoxicity/side effects of nanoparticles (generally or for a certain type of nanoparticles).
Author Response
We prepared a response letter to show our replies and the text we revised.

Reviewer 3 Report
In this review, the potential application of nanotechnologies to enhance the efficacy, quality and durability of drug delivery were investigated. For drug delivery in vivo, drug loaded in nanoplatforms target particular sites in a spatial- and tem-poral-dependent manner. The details were investigated in details. In my opinion, this work has good potential for publishing in this journal after some corrections:
1-Novel achievement of this work should be added in abstract section.
2-for better compression, it is need to graphical abstract.
3- Invitro study was missed.
4- The quality of Fig. 2 should be improved.
5-the interaction between experimental parameters were missed.
6- Recent advances about effects on nanotechnology in medicine should be added in text such as A: Design, Synthesis of Sulfadiazine Derivatives bearing Some New Heterocyclic Compounds with study Antimicrobial and Antioxidant Activity." Egyptian Journal of Chemistry (2022), B: Modification of Carbon Nanotubes Surface Using Different Oxidizing Agents. J Environ Anal Chem. 2015;2:e112, C: Asian Journal of Chemistry, 30(2), 460-462, D: Journal of Young Pharmacists 9, no. 4 (2017): 463. E: Molecular docking studies and biological evaluation of luteolin on cerebral ischemic reperfusion injury, Egyptian Journal of Chemistry, 65 (2022) 1-2.
7- English writing should be edit by native person
Author Response

(The authors gave the same response as above.)

Reviewer 4 Report
The authors prepared a review of the strategies for designing controlled drug delivery specifically in the scope of nanotechnologies. The authors also demonstrated good enough explanation with reasonable discussion but some problems that the authors should consider from the notes below.
1. Regarding the clearance in section 2.1, it would be better if the author added how long that the nanoparticles usually excrete from the body or the cells.
2. Please add the reason in detail why the size 10-200 nm, spherical NPs, and neutral nanoparticles become the best choice (section 2.1). It is quite an ambiguous sentence about “slightly negatives” and the word “anionic” comes after (Page 3 lines 110-111).
3. Please add an explanation in section 3.1.1 about the differences between the pH of some specific organs that explain here. It is better to add, strictly at what pH is it. Please also tell about the pH of ECM for this explanation (section 3.1.1)
4. In section 3.1.3, the author explains the photo-triggered release. Interestingly, the author also explains magnetic-stimulated nanoparticles. Based on the principle alone, both are different approaches. We suggest the author separate these two approaches.
5. In section 4.2, both covalent and non-covalent hydrogel is not well explained. Please improve this part.
6. Please check the typing, for example, Bi2Se3 should be Bi2Se, released at line 244, and the comma after esterases and stability (lines 270 and 302), should be written in italic (depending on journal writing format)
7. Please improve the English, some part is too long and complicated.
Author Response

(The authors gave the same response as above.)

Round 2
Reviewer 2 Report
I agree to publish in present form.
Reviewer 3 Report
Authors revised manuscript carefully. The final version can be published in this journal.